# The Role of Alternative Mitophagy in Heart Disease

**DOI:** 10.3390/ijms24076362

**Published:** 2023-03-28

**Authors:** Jihoon Nah

**Affiliations:** Department of Biochemistry, Chungbuk National University, Chungdae-ro 1, Seowon-gu, Cheongju-si 28644, Chungcheongbuk-do, Republic of Korea; jhnah@cbnu.ac.kr

**Keywords:** conventional autophagy, alternative autophagy, mitophagy, heart disease, Rab9

## Abstract

Autophagy is essential for maintaining cellular homeostasis through bulk degradation of subcellular constituents, including misfolded proteins and dysfunctional organelles. It is generally governed by the proteins Atg5 and Atg7, which are critical regulators of the conventional autophagy pathway. However, recent studies have identified an alternative Atg5/Atg7-independent pathway, i.e., Ulk1- and Rab9-mediated alternative autophagy. More intensive studies have identified its essential role in stress-induced mitochondrial autophagy, also known as mitophagy. Alternative mitophagy plays pathophysiological roles in heart diseases such as myocardial ischemia and pressure overload. Here, this review discusses the established and emerging mechanisms of alternative autophagy/mitophagy that can be applied in therapeutic interventions for heart disorders.

## 1. Introduction

The heart has a tremendous energy demand in the form of ATP for continuous myocyte contraction. As the cellular power plant, cardiac mitochondria provide the most ATP in the heart. While humans produce and consume approximately 65 kg of ATP daily, the human heart generates approximately 6 kg (8%) of the entire body’s total ATP [1]. Therefore, mitochondrial dysfunction is pivotal in heart diseases because of energy supply shortage and increased production of reactive oxidative species (ROS) under pathological conditions [2]. It can lead to cardiovascular diseases; hence, maintaining optimal mitochondrial homeostasis is essential in cardiomyocytes.

The quality control process is orchestrated by coordinating mitochondrial homeostasis, including fission and fusion, biogenesis, and degradation [3]. Mitochondrial dynamics are controlled by the fusion of outer mitochondrial membrane (OMM), facilitated by mitofusin 1 (Mfn1) and Mfn2, and inner mitochondrial membrane (IMM), aided by optic atrophy 1 (Opa1) protein, followed by fission into separate daughter mitochondria with the help of dynamin-related protein 1 (Drp1) [4]. The total mitochondrial content is mainly controlled by biogenesis and mitochondria-specific autophagy for degradation. Upregulation of proliferator-activated receptor-γ coactivator 1α activates the transcription of nuclear genes encoding mitochondrial proteins, including mitochondrial transcription factor A—the initial mechanism of mitochondrial biogenesis under appropriate stimuli [3]. Eventually, excessive or dysfunctional mitochondria are selectively recycled through a specialized autophagic pathway—mitophagy [5].

Autophagy is an intracellular catabolic process for cell degradation and recycling of cellular components, including proteins and organelles, in a lysosome-dependent manner. The pivotal role of autophagy is to maintain cellular homeostasis and function by eliminating dysfunctional proteins and organelles. The selective sequestration of cargo materials into a double-membrane structure, called the autophagosome, is governed by multiple autophagy-related (ATG) genes [6]. Although the most characterized mechanism of autophagy is the Atg5-, Atg7-, and LC3-mediated conventional autophagy pathway, recent studies have identified an Atg5/Atg7-independent pathway called alternative autophagy [7]. Instead of Atg5- and Atg7-dependent LC3 lipidation, alternative autophagy utilizes Rab9 to generate double-membrane autophagosomes [8]. The molecular mechanism and cellular function of alternative autophagy remain largely unexplored compared to the well-studied conventional autophagy pathway. The author and Dr. Sadoshima group recently discovered that the alternative mechanism is mainly involved in the elimination of damaged mitochondria in chronic heart conditions [8,9,10]. This review summarizes the molecular mechanisms as well as functional roles of mitophagy and discusses its pathophysiological significance in heart diseases.

## 2. Molecular Mechanism of Autophagy

### 2.1. Conventional Autophagy

A traditional topic in the autophagy field is understanding the molecular machinery of autophagosome biogenesis. Depending on the method of sequestering cargo materials, autophagy can be categorized as macro-, micro-, or chaperone-mediated autophagy. Macroautophagy—hereafter simply referred to as autophagy—segregates cytosolic cargo through a double-membraned structure called the autophagosome, which degrades the cargo by subsequent fusion with lysosomes. Over 40 Atg genes are involved in the conventional autophagy pathway [11]. Autophagy is generally induced in response to a change in extracellular nutrient availability, which is recognized by energy-sensing signaling pathways such as mammalian target of rapamycin complex 1 (mTORC1) and AMP-activated protein kinase (AMPK) signaling. Post-translational modifications of the ULK1 kinase complex (which consists of ULK1, Atg13, Atg101, and FIP200) induced by AMPK or mTORC1 are involved in initiating conventional autophagy. Class III phosphatidylinositol 3-kinase complex I (PI3KC3-C1) generates PI3P during the nucleation of phagophore, the primary double-membrane sequestering compartment [12]. This complex consists of Atg6/Beclin1, Atg14, PI3KC3/VPS34, and the regulatory subunit Vps15/p150. Vesicles of the sole multi-spanning membrane protein, Atg9, act as a membrane source for autophagosome nucleation and expansion [13]. The Atg18 and WIPIs proteins function as PI3P effectors in autophagosome formation, transmitting PI3P signals to the downstream ATG proteins [13]. A prominent marker protein for conventional autophagy is Atg8/LC3. The newly translated LC3 (pro-LC3) proteins are immediately processed at the C-terminus by Atg4 protease, leading to LC3-I formation [14]. Two ubiquitin-like systems, Atg12 and Atg8/LC3 conjugation, essentially act as E3-like systems in LC3-PE (LC3-II) conjugation [15]. Lastly, the double-membrane autophagosomes mature by completing their expansion, thereafter facilitating SNARE-mediated fusion with lysosomes to eliminate the cargo materials. The brief molecular mechanism of conventional autophagy is described in Figure 1A.

### 2.2. Discovery of Atg5- and Atg7-Independent Alternative Autophagy

The mechanisms of molecular autophagy orchestrated by multiple ATG genes have been extensively studied. These investigations demonstrated that *Atg5* and *At*g7 are essential for mammalian autophagy and that LC3 lipidation is a master indicator for this process. However, interestingly, Nishida et al. identified an alternative *Atg5*- and *Atg7*-independent pathway, hereafter referred to as alternative autophagy, under an electron microscope using Atg5 knockout cells [7]. Alternative autophagy does not utilize the two ubiquitin-like conjugation systems that involve Atg5, Atg7, and LC3. Nonetheless, both conventional and alternative autophagy have similar morphological and functional features and share some Atg genes, such as *Ulk1*. Similar to conventional autophagy, alternative autophagy leads to the formation of double-membrane structures, i.e., autophagosomes, which fuse with lysosomes to degrade the internal cargoes. However, unlike conventional autophagosomes, alternative autophagosomes are generated in a Rab9-dependent manner rather than LC3 [7] (Figure 1B).

Owing to the limitation of indicators and assays, it is still challenging to unveil the complete molecular mechanisms of *Atg5*- and *Atg7*-independent alternative autophagy. To date, only a few additional processes have been identified. The *Ulk1* protein is the mammalian ortholog of yeast *Atg1*. Two paralogs, *Ulk1* and *Ulk2*, are involved in initiating conventional autophagy. However, *Ulk2* is considered redundant, whereas loss of *Ulk1* is generally sufficient to disrupt the process [16]. Unlike *Ulk2*, which is only involved in conventional autophagy, *Ulk1* is an essential initiator of both conventional and alternative processes [17]. Phosphorylation of *Ulk1* at multiple sites by mTORC1 or AMPK prevents or activates both conventional and alternative autophagy [18]. In addition, activated *Ulk1* phosphorylates several autophagy-related targets, including Beclin1 and VPS34 [19]. A recent study by Torii et al. identified the mechanism underlying the *Ulk1*-mediated regulation of alternative autophagy [20]. Torii et al. first identified Ulk1 phosphorylation sites in response to potent alternative autophagy activation by etoposide treatment. Phosphorylation of Ulk1 at Ser746 is a critical modulator of alternative autophagy in etoposide-treated Atg5-knockout MEF cells. In addition, an immunoprecipitation assay revealed an interesting interaction between Ulk1 and the necroptosis-regulating kinase RIPK3 under alternative autophagy-activating conditions. The RIPK3-dependent phosphorylation of Ulk1 at Ser746 activates only alternative and not conventional autophagy or necroptosis, which indicates a distinct functional difference in Ulk1 between conventional and alternative autophagy [20,21].

### 2.3. Comparison of Conventional and Alternative Autophagy

The apparent differences between conventional and alternative autophagy are the status of Ulk1 and the source of autophagosomes. In conventional autophagy, Ulk1 complexes with Atg13, FIP200, and Atg101. Interestingly, this complex can dissociate under alternative autophagy-inducing conditions, accompanied by Ulk1 phosphorylation at Ser746. Free Ulk1, dissociated from the Ulk1/FIP200/Atg13/Atg101 complex, translocates to the Golgi membrane to facilitate alternative autophagy [20]. However, it remains unclear whether Ulk1 recruits other target proteins. Its phosphorylation status is the key difference between conventional and alternative autophagy. Under physiological conditions, Ulk1 is phosphorylated by mTORC1 at Ser638 (corresponding to Ser637 of mouse Ulk1) and Ser757 to inactivate autophagy [22,23]. Under autophagy-inducing conditions, Ser638 and Ser737 are dephosphorylated by several phosphatases, whereas sites such as Ser317 and Ser777 are phosphorylated by AMPK to stimulate autophagy [18]. Interestingly, Ser638 dephosphorylation is crucial in both conventional and alternative autophagy. Studies have revealed that the phosphorylation status of Ulk1 at Ser638 decreases upon starvation. However, recent studies have uncovered that Ser638 dephosphorylation by protein phosphatase 1D magnesium-dependent delta isoform is necessary for genotoxic-induced alternative autophagy [24]. Whether and how Ulk1 dephosphorylation at Ser638 modulates conventional or alternative autophagy requires further investigation. Moreover, Ulk1 phosphorylation at Ser747 (corresponding to Ser746 of mouse Ulk1) has recently been identified as an alternative autophagy-specific phosphorylation mechanism under genotoxic stress. The protein kinase RIPK3 facilitates this phosphorylation following the dephosphorylation at Ser638 [20]. Interestingly, unlike other phosphorylation sites conserved in both Ulk1 and Ulk2, the two important sites in alternative autophagy, Ser 638 and Ser747, are conserved in Ulk1 but not in Ulk2 in higher vertebrates.

Another difference is the membrane source of autophagosomes. Conventional autophagosomes originate from multiple intracellular membranes, including those from the endoplasmic reticulum (ER), mitochondria, and ER-mitochondria contact site, along with the plasma membrane [25], whereas the alternative autophagosomes originate from the trans-Golgi network [7]. Therefore, alternative autophagy-inducing conditions, such as etoposide, amphotericin B1, and genotoxic stress, may be directly or indirectly involved in Golgi-mediated stress [20,26,27]. As Atg5/Atg7-dependent LC3 lipidation is not associated with the generation of alternative autophagosomes, the orchestration of this process remains unclear. Alternative autophagosomes are produced in an Rab9-dependent manner and mediate the elongation of isolation membranes with vesicles derived from the trans-Golgi and late endosomes [7]. Recent studies have identified candidates involved in modulating alternative autophagosome generation. As mentioned above, the WD-repeat protein interacting with the phosphoinositide (WIPI) family functions as an essential PtdIns3P effector at the isolation membrane [28]. Unlike Wipi1 and Wipi2, which play vital roles in conventional autophagy, Wipi3 binds to the Golgi membrane, which is essential for generating alternative isolation membranes. Nonetheless, it plays a minor role in conventional pathways [27]. In addition, another recent report revealed that Golgi-resident Rab2 participates in autophagosome formation by dissociating from the Golgi apparatus under autophagy-inducing conditions to interact with Ulk1 [29]. However, this report did not directly reveal its role in supplying the Golgi membrane for autophagosome generation in alternative autophagy, which requires further investigation.

Unlike mice with knockouts of many essential *Atg* genes, including *Atg5*, *Atg7*, and *Atg12*, that are lethal to neonates, *Ulk1*-deficient mice are viable [30]. However, *Ulk1*-knockout mice display defects in mitochondrial autophagy during erythrocyte development and in primary hepatocytes [17,22]. Based on the Ulk1 deficiency characteristics seen in alternative autophagy, which differ from those in conventional autophagy, understanding the role of the alternative pathway in mitochondrial homeostasis under stress conditions is noteworthy. Accordingly, in Section 2, the functional importance of alternative autophagy pathways in mitophagy is described.

## 3. Diverse Molecular Mechanisms and Functions of Mitophagy

The double-membraned organelle mitochondria are responsible for producing ATP and execution of programmed cell death. Two membranes can divide the mitochondrial compartments. The outer mitochondrial membrane separates the mitochondrial compartment from the cytosolic area of the cell. The inner membrane of the mitochondria is highly folded, forming structures called cristae. These folds increase the surface area of the inner membrane for more efficient ATP production. The space within the inner membrane is known as the mitochondrial matrix, which contains mitochondrial DNA [31]. Mitochondria are highly dynamic organelles that constantly undergo homeostatic processes such as biogenesis, fission, fusion, and degradation [2]. During energy metabolism, overproduced ROS can damage the mitochondria, which should be dealt with immediately. Mitochondrial fusion may dilute defective components. However, mitophagy, which occurs with mitochondrial fission, is essential in eliminating the damaged portions of mitochondria [32]. Based on its labeling of the damaged portion that should be destroyed, mitophagy can be categorized as PINK1/Parkin-mediated or receptor-dependent pathway.

### 3.1. Ubiquitin-Mediated Mitophagy

PINK1 is a serine/threonine kinase that is degraded inside the mitochondrial matrix by matrix-processing peptidase and presenilin-associated rhomboid-like (PARL) protease. It is then retro-translocated to the cytosol for further degradation in healthy mitochondria [33]. PINK1 accumulates in the outer mitochondrial membrane via multiple cellular processes in response to depolarization of the mitochondrial membrane potential. Upon mitochondrial ATP depletion, PARL becomes inactivated because of N-terminal autocatalytic cleavage, causing PINK1 accumulation [34]. In damaged mitochondria, the translocase of the outer membrane (TOM) complex does not import PINK1. Therefore, it remains at the OMM and is stabilized and autophosphorylated [35]. Loss of mitochondrial membrane potential is the key trigger for PINK1 accumulation in OMM. However, recent studies have identified additional supporting factors. Without Tom7, a small accessory protein of the TOM complex, PINK1 cannot accumulate in the OMM, even in depolarized mitochondria. Tom7 is essential for PINK1 accumulation in OMM due to mitochondrial membrane potential loss [36]. The CHCHD4/GFER disulfide relay system, a mammalian homolog of the yeast Mia40/Erv1 redox system that transfers proteins into the mitochondrial intermembrane space, is required for PINK1 stabilization through protein–protein interactions in the OMM [37]. The ROS-induced mitochondrial dysfunction increases CHCHD4/GFER activity, thereby promoting PINK1 accumulation [38].

Parkin is a RING-between-RING E3 ubiquitin ligase that functions in the covalent attachment of ubiquitin to specific substrates. Consistent with its name, the mutations in Parkin are closely linked to Parkinson’s disease, as well as a few other human diseases, such as cancer and mycobacterial infection [39]. A structural investigation revealed that Parkin exhibits a low basal activity due to intramolecular interactions that block its E2-binding site [40]. Upon mitochondrial damage, PINK1 recruits Parkin to the OMM and activates it through either ubiquitin Ser65 or Parkin Ser65 phosphorylation, which induces an active conformational change in the E2-binding site of Parkin [41,42,43]. Studies on cardiomyocytes have revealed Mfn2 as another PINK1 substrate. PINK1 phosphorylates Mfn2 at S442 and T111, recruiting Parkin to damaged mitochondria [44]. Once fully activated in the damaged mitochondrial membrane, the E3 ubiquitin ligase Parkin ubiquitinates various target proteins in two stages: OMM and mitochondrial matrix proteins in the early stage, and IMM proteins in the late stage [45].

Parkin-mediated poly-ubiquitination of mitochondrial proteins in the OMM recruits mitophagy adaptor proteins (LC3-anchor proteins) such as optineurin (OPTN), NDP52, Tax1 binding protein 1, NBR1, and p62 (Figure 2A) [2]. Mitophagy adaptors harbor a ubiquitin-binding domain and an LC3-interacting region (LIR) associated with poly-ubiquitin-coated damaged mitochondria, concomitantly recruiting LC3 to promote autophagosome formation. The initial pathway of PINK1/Parkin-mediated ubiquitination is well established. Nonetheless, clarification on how autophagosomes are generated to sequester ubiquitinated mitochondria is still required. Mitophagy adaptors may be essential in recruiting LC3 and upstream autophagy-related proteins, including Ulk1 and WIPI1, to generate autophagosomes [46]. NDP52 interacts with the Ulk1 complex via FIP200, a critical component of the Ulk1 complex. NDP52 phosphorylation by TBK1 in depolarized mitochondria enhances NDP52 and FIP200 binding. Interestingly, NDP52 can activate Ulk1 independent of the AMPK and mTOR status [47]. OPTN complexes with Atg9A vesicles, and a disruption in this interaction causes mitophagy dysfunction [48]. In addition to that in the mitochondrial membrane, OPTN and Atg9A colocalization has also been detected near foci in the perinuclear region [49]. This might indicate the dichotomous role of OPTN in recruiting LC3 and upstream autophagy regulators.

### 3.2. Receptor-Dependent Mitophagy

The involvement of mitochondrial receptors and adaptors is essential for mitophagy initiation. Unlike mitochondrial adaptors, which recognize the PINK1/Parkin-mediated poly-ubiquitination of mitochondrial proteins in OMM, mitophagy receptors do not require polyubiquitination to enhance the process. Instead, damaged mitochondria are directly targeted by LC3-dependent autophagosomes via mitophagy receptors with a LIR [4,50]. Several mitophagy receptors, including FUN14 domain-containing protein 1 (FUNDC1), BNIP3, NIX, Prohibitin2, and cardiolipin, have been characterized (Figure 2B). These receptors are localized in the OMM and inner mitochondrial membrane [51].

BNIP3L/NIX is the most thoroughly studied example of a mitophagy receptor, which was initially known to clear the mitochondria during reticulocyte development [52]. BNIP3L is a BH3-only protein localized to the OMM and interacts with pro-apoptotic Bcl2 family proteins such as BAX and BAK [53]. Interestingly, BNIP3L disrupts the Bcl2-Beclin1 interaction, releasing Beclin1 to promote autophagy [54]. In addition, BNIP3L interacts with Atg8 family proteins through its LIR motif, which is essential for enhancing mitophagy [55]. Phosphorylation near the WVEL LIR motif of BNIP3L at Ser34 and Ser35 is essential for fine-tuning the BNIP3L-LC3 interaction and initiating mitophagy [56]. Similar to other BH3 proteins, BNIP3L interacts to form dimers on the OMM, which is essential for robustly recruiting autophagy-regulating components to the mitochondria [57]. However, in some cases, BNIP3L participates in ubiquitin-mediated mitophagy as a Parkin substrate on the depolarized mitochondrial membrane. Parkin-mediated poly-ubiquitination of BNIP3L results in recruitment of the mitophagy adaptor NBR1, which targets mitochondria for degradation [58].

FUNDC1 is an integral OMM protein that is essential for the regulation of mitophagy in response to hypoxic conditions. The stability of FUNDC1 is tightly regulated by the mitochondrial E3 ubiquitin ligase MARCH5/MITOL. FUNDC1 degradation via ubiquitylation at lysine 119 desensitizes the mitochondria to hypoxia-induced mitophagy, indicating the functional importance of FUNDC1 as a mitophagy receptor in such conditions [59]. Thus, protein stability and FUNDC1 phosphorylation are critical in hypoxia-induced mitophagy. Similar to that with BINP3L, the FUNDC1–LC3 interaction is regulated by phosphorylation near its LIR motif. Two phosphorylation sites near the LIR motif—Ser13 and Tyr18—are phosphorylated by CK2 and Src, respectively, under basal conditions and interfere with the FUNDC1 and LC3 interaction [60,61]. A structural analysis revealed that the third LIR residue (Val20) of FUNDC1 is unusually inserted into the hydrophobic pocket of LC3 [62]. Consequently, Tyr18 phosphorylation weakens the FUNDC1-LC3 binding affinity via electrostatic repulsion. However, Ser17 phosphorylation by Ulk1 near the LIR motif stabilizes the FUNDC1–LC3 interaction [63].

Unlike most mitophagy receptors in the OMM, an inner mitochondrial membrane protein, prohibitin 2 (PHB2), can function as a mitophagy receptor. Interestingly, the PHB2 and LC3 interaction requires Parkin-mediated poly-ubiquitination [64]. The OMM is a major hurdle in the PHB2 and LC3 interaction. Wei et al. demonstrated that Parkin-mediated poly-ubiquitination of OMM proteins causes small ruptures outside the mitochondria because of proteasome-mediated degradation of target proteins, providing physical space for the PHB2-LC3 interaction [64]. A recent study suggested another role of PHB2 in mitophagy—PHB2 mediates PINK1 stabilization by negatively modulating PARL activity, thereby regulating Parkin/PINK1-mediated mitophagy regardless of its LC3 binding [65].

### 3.3. Rab9-Dependent Alternative Mitophagy

As mentioned above, research on mitophagy has revealed a complicated molecular mechanism for the selective destruction of damaged mitochondria. However, most studies have intensively investigated Parkin-mediated ubiquitination and receptor-mediated pathways, which directly or indirectly interact with LC3 through the LIR motif. Although Atg5/Atg7-independent autophagy was identified in 2009 [7], the molecular mechanism by which it carries out mitochondrial clearance remains largely unknown. Technically, it is challenging to discover an alternative mechanism for LC3-independent, Rab9-dependent mitophagy. Early mitophagy studies utilized fluorescence microscopy to examine the colocalization of LC3-mediated autophagosomes or lysosomes with mitochondria or electron microscopy to detect the ultrastructure of degrading mitochondria directly [65]. However, mitophagy research has become more diverse and active with improvements in technique. For instance, several novel assays, such as MitoTimer, Mito-Keima, and Mito-QC, have been recently developed to detect mitophagy [66]. Therefore, the understanding of LC3-independent mitophagy has expanded.

Dr. Mizushima’s group reported early observations of alternative mitophagy. LC3-mediated autophagosome generation is essential for efficiently incorporating damaged mitochondria in Parkin-mediated mitophagy. Nevertheless, structures containing the Atg9A and Ulk1 complex can associate with depolarized mitochondria, even without the membrane-bound LC3 [67]. During erythrocyte maturation, erythroblasts lose their organelles, including nuclei and mitochondria. Ulk1 is vital in this process; however, the roles of Atg5 and Atg7 are controversial [68]. The pivotal role of the Ulk1/Rab9-dependent alternative pathway in clearing mitochondria during erythrocyte maturation has been revealed using electron microscopy in Ulk1-deficient and Atg5/Ulk1-deficient mice [69]. The alternative mitophagy pathway is also essential in the induced pluripotent stem cell (iPSC) approach. Metabolic reprogramming from mitochondrial oxidative phosphorylation to glycolysis is involved in iPSC reprogramming. Interestingly, autophagic processes that are Atg5-independent but Ulk1- and Rab9-dependent mediate mitochondrial clearance while manipulating cellular plasticity. Blockage of alternative autophagy inhibits mitochondrial clearance, thereby preventing iPSC induction [70].

Hirota et al. raised a question regarding the role of Parkin/PINK1 in physiological mitophagy, as researchers often need to overexpress Parkin in certain cell types for mitophagy induction [71,72]. Parkin and PINK1 are related to certain types of mitophagy, such as that induced by carbonyl cyanide m-chlorophenyl hydrazone (CCCP), which is a typical method of induction in vitro; however, it does not occur in physiological conditions [5,72,73]. Instead of CCCP-treatment, Hirota et al. examined starvation- or hypoxia-induced mitophagy in HeLa cells using the pH-sensitive fluorescent protein Keima (Mito-Keima). Interestingly, it was found to be largely Atg5/Atg7-independent but Ulk1/Beclin1/Rab9-dependent [72].

The functions of mitochondria are maintained by their dynamics—fission and fusion. Mitochondrial fission facilitates mitophagy by separating the damaged mitochondrial portion from the healthy one [74]. The separated depolarized portion of mitochondria is prevented from re-fusing with the healthy part by the decreased level of fusion protein OPA1 [75]. Dynamin-related protein 1 (Drp1) is recruited to the point of mitochondrial fission via mitochondrial fission factor and oligomerizes on the OMM in a ring-like structure to divide the mitochondria via mechanical forces [76]. Drp1 is an essential protein in various mitophagy types, including Parkin-dependent, Parkin-independent, receptor-mediated, and alternative mitophagy [77,78]. The activity of Drp1 is modulated by post-translational modifications, particularly phosphorylation. Mitochondrial fission is enhanced through Drp1 phosphorylation at Ser616 by multiple kinases, such as Cdk1, Erk2, and Rip1, depending on the situation [8]. In contrast, phosphorylation at Ser637 by cAMP-dependent protein kinase reduces the GTPase activity and inhibits mitochondrial fission in cells [79]. Drp1 and Parkin play a synergistic role in heart and brain mitophagy. Loss of Drp1 shows embryonic lethality due to abnormal forebrain development [80], and cardiomyocytes-specific Drp1 deficiency causes fatal cardiac dysfunction by inhibiting mitophagy, independent of Parkin [77]. In addition, depletion of Drp1 in cardiomyocytes causes mtDNA nucleoid clustering and malfunction mitochondrial respiration [81].Recent studies have revealed that Drp1 phosphorylation in the heart promotes alternative mitophagy. The Ulk1-mediated phosphorylation of Rab9 at Ser179 is an essential initial step for alternative mitophagy under stressful conditions [8]. Activated Ulk1 and phosphorylated Rab9 are platforms for the formation of a signaling complex between Rip1 and Drp1. Mitochondrial fission is promoted by Rip1-mediated Drp1 phosphorylation at Ser616, and Rab9-mediated autophagosomes are generated around dysfunctional mitochondria [8] (Figure 3). It is worth noting that although Drp1 plays an essential role in mitochondrial fission, Drp1-independent mitochondrial division might occur during mitophagy in some cases [82].

The role of the association of mitochondrial dynamics—other than that of Drp1—in regulating alternative mitophagy has yet to be understood completely. Mitophagy adapters and receptors interact with LC3 through the LIR motif. However, some studies have also identified an LC3-independent role of these proteins. As mentioned above, the mitophagy adapter NDP52 recruits the Ulk1/FIP200 complex to ubiquitinated mitochondria through an LC3-independent mechanism [47]. Another mitophagy receptor, OPTN, strongly correlates with Atg9A, even under Atg5-deficient conditions [48]. Interestingly, the OPTN-Atg9A foci are found in the Golgi complex, a vital membrane source for alternative mitophagy [49]. Under conditions of deficient conventional autophagy after Atg7 or FIP200 inactivation, alternative forms of mitochondrial quality control compensate for the LC3-dependent mitophagy in cancer cells. The approximate 50-nm diameter mitochondrial-derived vesicles deliver a minimal portion of mitochondrial material to a lysosome in an LC3-independent manner. This mechanism is regulated by the BAR-domain-containing SNX9 [83]. Interestingly, SNX18, a paralog of SNX9, regulates Atg9A vesicle trafficking in the ER and Golgi apparatus [84]. These observations imply the potential role of proteins related to mitochondrial dynamics in regulating alternative mitophagy.

## 4. Functional Role of Alternative Mitophagy in Heart Diseases

Various mitophagy-regulating pathways have been identified in response to different types of stress. The utility of modulating the signaling pathways that promote mitophagy should be quantified and validated for developing therapeutic approaches for heart diseases. Newly developed mitophagy assays, especially those using Mito-Keima transgenic mice, help to monitor and quantify mitophagy in vivo [85]. In addition, Dr. Sadoshima’s group recently identified the role of alternative mitophagy in various heart disorders using a mouse cardiac-specific Mito-Keima system (Figure 4) [8,9,10].

### 4.1. Ischemic Cardiomyopathy

Myocardial infarction (MI), generally known as a heart attack, occurs because of the reduction or stoppage of blood flow in a major branch of the coronary artery of the heart. It is a leading cause of death and disability worldwide [86]. Insufficient blood supply deprives the cardiomyocytes of oxygen and nutrients, which then activate autophagy to maintain energy balance by recovering ATP generation. Mitochondrial dysfunction is commonly observed during MI and is a critical determinant of cardiomyocyte death. Therefore, preventing mitochondrial dysfunction is an essential therapeutic strategy for cardioprotection during MI [87]. Mitophagy increases during MI through various signaling pathways, including Parkin- and receptor-mediated pathways [61,88,89]. However, the role of alternative mitophagy had not been identified until recently by Saito et al., who clearly distinguished the roles of conventional and alternative mitophagy in MI [8]. Cardiac-specific Atg7 KO mice (*atg7*cKO), a conventional autophagy-deficient model, displayed almost complete mitophagy inhibition at baseline, and starvation or ischemia-induced mitophagy was preserved, similar to that in the control mice. Similar to *atg7*cKO, Parkin KO also exhibits preserved mitophagy activation under energy stress. Contrarily, *ulk1*cKO, a mouse model with a preserved alternative pathway but deficient in conventional autophagy, displayed a drastic reduction in mitophagy under energy stress conditions [8]. These results suggest that the conventional autophagy pathway helps maintain mitochondrial function under basal conditions. In response to energy stress, including ischemia, cardiomyocytes stimulate alternative mitophagy, rather than conventional or Parkin-dependent mitophagy, to preserve mitochondrial function. In addition, brefeldin A, a Golgi-derived membrane inhibitor, notably inhibits only alternative mitophagy in response to energy stress, which confirms that the Golgi membrane is an essential source of alternative autophagosomes [8]. During energy stress, mitophagy is activated by AMPK-mediated Ulk1 phosphorylation, which is the driving force of further signaling cascades, including Rab9 phosphorylation at Ser179 and Drp1 phosphorylation at Ser616. Mass spectrometry revealed that a large Ulk1/Rab9/Rip1/Drp1 complex facilitates Parkin-independent alternative mitophagy under heart energy stress. The importance of Rab9 phosphorylation by Ulk1 in alternative mitophagy was revealed in phosphorylation-defective knock-in mice (Rab9-S179A). The Rab9-S179A mice exhibit severely impaired mitophagy during ischemia, with an enlarged infarction area [8].

### 4.2. Hypertrophic Cardiomyopathy and Heart Failure

Cardiac hypertrophy is an adaptive response to pressure or volume overload. A hypertrophic heart undergoes abnormal enlargement or thickening of the wall to normalize diastolic pressure [90]. An initial study revealed that the LC3-dependent conventional autophagy rapidly increases half a day after pressure overload induced by transverse aortic constriction (TAC). This is quickly attenuated under physiological levels. However, using electron microscopy and Mito-Keima, mitophagy was found to be activated 3–7 days post-TAC [91]. The unmatched activation times between LC3-dependent autophagy and mitophagy during pressure overload indicate that mitophagy may not depend on the conventional pathway. A recent report identified Ulk1/Rab9-mediated alternative mitophagy as being predominant, occurring during pressure overload [9]. TAC transiently induced LC3-dependent conventional autophagy early within a single day. Subsequently, Ulk1/Rab9-mediated alternative mitophagy gradually increased and peaked 3–5 days after TAC. A comparative mitophagy assay using Mito-Keima mice revealed that alternative mitophagy is considerably more dominant than the conventional form during pressure overload. Heart failure generally appears two weeks post-TAC, whereas defective alternative mitophagy causes early cardiac dysfunction three days post-TAC [9]. LC3-dependent autophagy is temporarily induced one day after TAC. Nonetheless, the physiological importance of conventional autophagy in heart failure is also an adaptive response. Tamoxifen-induced Atg5 knockout in cardiomyocytes results in severe cardiac dysfunction, even under basal conditions. In addition, cardiac-specific Atg5 knockout mice display early cardiac dysfunction one week post-TAC [92]. The author recently demonstrated that a potent autophagy-inducing peptide, TAT-Beclin1, promotes conventional and alternative mitophagy in cardiomyocytes. Interestingly, the TAT-Beclin1-induced mitophagy rescues early cardiac dysfunction of *ulk1*cKO in response to pressure overload [9]. These results suggest that the prolonged LC3-dependent mitophagy activation can compensate for alternative mitophagy function during heart pressure overload. Understanding how different mitophagy forms are sequentially activated and the different roles of conventional as well as alternative mitophagy during pressure overload is essential.

### 4.3. Diabetic Cardiomyopathy

Diabetic cardiomyopathy is characterized by cardiac dysfunction and abnormal myocardial structure in patients with diabetes; it occurs in the absence of other cardiovascular diseases [93]. Under diabetic conditions, mitochondria switch ATP sources from glucose to fatty acid oxidation because of reduced insulin activity; thereby, mitochondrial function worsens because of increased ROS production [94]. Decreased mitophagy in chronic diabetes could be a critical cause of diabetic cardiomyopathy [95]. Autophagy and mitophagy regulation in diabetic hearts is complicated, and the results are incongruent. Cardiac autophagy and mitophagy can be unaltered, activated, or diminished, depending on the stage and severity of diabetes [6]. Some studies have reported that autophagy declines during the chronic phase of type 2 diabetes, while others have indicated the opposite [96,97]. A recent study by Tong et al. observed that the LC3-dependent cardiac autophagy in mice fed with a high-fat diet (HFD) peaked at six weeks and declined thereafter. In contrast, mitophagy is continuously activated for two months [98]. These results suggest that alternative mitophagy could be activated in the chronic phase of HFD-induced diabetic hearts. Soon after, the same group reported that an alternative Ulk1/Rab9 pathway mediates mitophagy during the chronic HFD feeding phase. Alternative mitophagy is activated, probably as a compensatory mechanism. On the other hand, conventional mitophagy is inactivated during the chronic phase of HFD feeding, during which the TFE3-dependent transcriptional activation of Rab9 promotes alternative mitophagy [10]. Interestingly, TFE3 is a member of the Tfeb family and a master transcription factor of autophagy and lysosomal genes [99]. However, the mechanism by which Tfeb and TFE3 regulate conventional and alternative autophagy in coordination remains unclear. In addition to TAT-Beclin1, Rab9 overexpression promotes alternative mitophagy and protects the heart against HFD-induced cardiac dysfunction. Upregulation of alternative mitophagy by Rab9 overexpression rescues mitophagy and cardiac function even in *atg7*cKO mice in response to HFD feeding. This indicates that alternative mitophagy can compensate for the functional role of conventional mitophagy [10].

## 5. Therapeutic Perspective of Alternative Autophagy

Preserving or increasing the mitochondrial function is an attractive target for clinical intervention in various heart diseases as mitochondria play an essential role in the heart and dysfunctional mitochondria lead to cardiac dysfunction [100]. Recently, strategies for recovering mitochondrial function have been widely developed. For instance, elamipretide—a water-soluble, aromatic-cationic, mitochondria-targeting peptide—associates with cardiolipin to rescue mitochondrial bioenergetics [101]. A recent report showed that elamipretide promotes engulfment of mitochondria into autophagosomes even under nutrient rich conditions [102]. In addition, the cell-to-cell mitochondrial transfer and ex vivo transplantation have been studied to replace dysfunctional mitochondria with new functional ones [103,104]. Together with these methods, mitophagy activation has been extensively explored. Many medications, including rapamycin, sulfaphenazole, statins, metformin, and resveratrol, have been introduced as effective autophagy inducers in cardiovascular diseases [105]. Although no drugs that selectively activate mitophagy have yet been discovered, mitophagy is generally activated by many autophagy inducers. Spermidine is a polyamine in our diet that rapidly activates autophagy by inhibiting the acetyl transferase EP300 or activating Tfeb signaling [106,107]. Spermidine improves cardiac function by activating LC3-dependent autophagy and mitophagy in old hearts [108]. Urolithin A, a gut metabolite, also induces LC3-dependent autophagy and mitophagy in *C. elegans* and human skeletal muscles [109,110]. TAT-Beclin1, a cell-penetrating potent autophagy-inducing peptide that interacts with the autophagy suppressor GAPR-1, is one of the most intensively investigated autophagy and mitophagy inducers in heart diseases. TAT-Beclin1 in vivo injection improves cardiac function in various heart disorders, including pressure overload, diabetic cardiomyopathy, and cardiac dysfunction by sepsis [9,10,111]. TAT-Beclin1 is also an alternative mitophagy inducer [9]. Beclin1 modulates conventional and alternative autophagy pathways, and inducing autophagy by targeting Beclin1 could stimulate both pathways [72]. Alternative mitophagy research is currently in its infancy; thus, effective interventions for targeting alternative mitophagy in heart diseases still need elucidation. Understanding the detailed molecular mechanisms and different roles of conventional and alternative mitophagy could be a promising approach for developing novel treatments for heart disease.

## 6. Conclusions

The functional role of mitophagy in cardiovascular disease is complex. Additional regulators besides those mentioned in this study may modulate the conventional and alternative mitophagy signaling pathways in cardiac disorders. Notably, the pathophysiological function of mitochondrial homeostasis, especially mitophagy, varies depending on the severity, duration, and region of cardiac stress. As mentioned above, the disease progression stage may be the primary determinant of the mitophagy-activating pathway between conventional and alternative methods. However, the physiologically appropriate point of alteration from conventional to alternative and the functional difference between both pathways remain unclear. Further elucidation of the roles of conventional and alternative mitophagy in maintaining heart function and the various correlations between both pathways would significantly aid in improving therapeutic strategies against cardiovascular diseases. Lastly, alternative autophagy plays an essential role in heart diseases, as mentioned in this review, and could be critical in various human diseases [112]. Thereby, understanding the precise role of alternative autophagy will ultimately be applied to various human diseases, including heart disease.

## Figures and Tables

**Figure 1 ijms-24-06362-f001:**
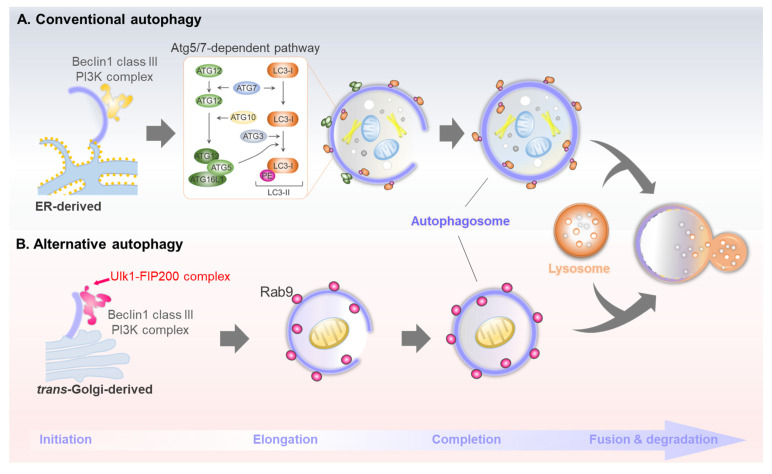
Brief molecular mechanisms of conventional and alternative autophagy. (**A**) Conventional autophagy is initiated by Ulk1/2, Beclin1 complex, and Atg5/7-dependent LC3 lipidation. (**B**) Alternative autophagy is initiated by Ulk1, Beclin1 complex, and Rab9-dependent autophagosome generation.

**Figure 2 ijms-24-06362-f002:**
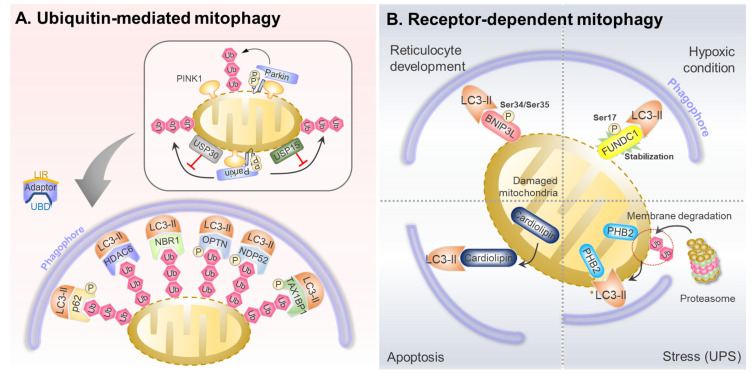
Receptors and adaptors of mitophagy. (**A**) Mitophagy adaptors have both LIR and UBD that interact with ubiquitin and LC3, respectively, in response to mitophagy activation. (**B**) Several mitophagy receptors facilitate mitophagy in response to various stress conditions.

**Figure 3 ijms-24-06362-f003:**
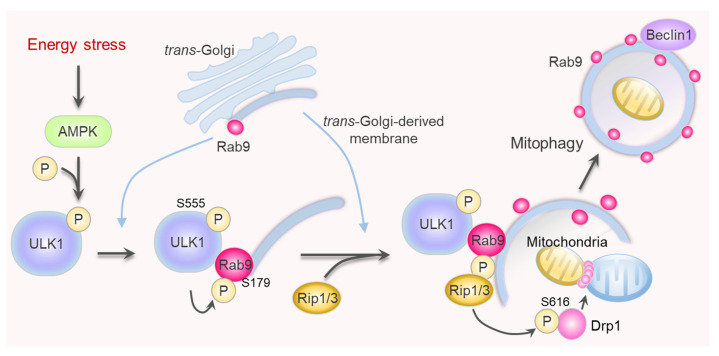
Molecular mechanism of Ulk1/Rab9-dependent alternative mitophagy. Upon mitophagy activation signaling, activated Ulk1 phosphorylates Rab9 at Ser179, thereby recruiting Rip1 and Drp1 to promote alternative mitophagy.

**Figure 4 ijms-24-06362-f004:**
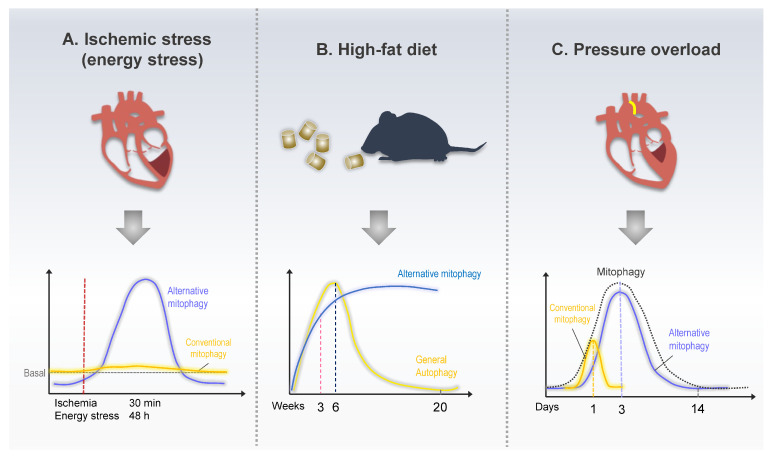
Schematic model of mitophagy in response to heart disease. (**A**) Mice exhibited increased alternative mitophagy levels 30 min after ischemia or 48 h after starvation, whereas conventional mitophagy did not change in response to energy stress. (**B**) Mice fed a high-fat diet (HFD) exhibited increased conventional mitophagy up to 6 w after the beginning of HFD feeding. The level of alternative mitophagy kept increasing even after 24 weeks of HFD feeding. (**C**). During pressure overload, conventional mitophagy was acutely activated within a day of TAC, whereas alternative mitophagy exhibited prolonged activation between 3 and 7 days after TAC.

## Data Availability

Not applicable.

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
