# Peer review of "The Role of Alternative Mitophagy in Heart Disease"

_ijms, 2023, doi:10.3390/ijms24076362_

Round 1

Reviewer 1 Report

Jihoon Nah has written a concise yet detailed review of the role of alternative mitophagy as a viable therapeutic target, especially in the context of heart disease.  Overall, this is a well-written and knowledgeable treatment of the topic, which will be a valuable resource for scientists interested in autophagy, in general, and mitophagy, specifically.  The main shortcoming of this article is omission of various key references, which should be discussed to present a more comprehensive picture.  If these points can be addressed, this review should be published.

Major comments:

Lines 37-38:  recent studies indicate that Fis1 may not promote fission but rather functions as an inhibitor of the fusion machinery (https://www.embopress.org/doi/full/10.15252/embj.201899748)

Lines 332-340: to add to this discussion about the physiological role of Pink1/Parkin please discuss Lee et al., 2018 (https://rupress.org/jcb/article/217/5/1613/38895/Basal-mitophagy-is-widespread-in-Drosophila-but)

Lines 341-343:  Twig et al., 2008 should be cited here (https://www.embopress.org/doi/full/10.1038/sj.emboj.7601963).  This article would benefit from a further description of how the mitochondrial life cycle is related to the turnover of damaged mitochondria.  An important detail about the role of mitochondrial dynamics in autophagy is that depolarized mitochondria are typically prevented from re-fusing with the mitochondrial network due to a loss of OPA1.

Line 343-344:  DRP1 is recruited via Mitochondrial fission factor (MFF).

Line 354:  Please cite and discuss Ishihara et al., 2015 (https://journals.asm.org/doi/full/10.1128/MCB.01054-14).

Line 352:  This line is lacking a citation.  Here or elsewhere, it would be appropriate to note the essentiality of DRP1 for neuronal development (https://www.nature.com/articles/ncb1907).

Line 367-368:  Yamashita et al., 2016 is an important article in this regard:  https://doi.org/10.1083/jcb.201605093

Line 497-499: Indeed there is evidence that Elamipretide promotes mitophagy, as revealed by Mito-QC experiments in this article:  https://www.sciencedirect.com/science/article/pii/S0022283618312348

Minor comments:

Line 31:  mitochondrial dysfunction underlies the pathophysiology of multiple tissues and cell types, e.g., optic neuropathy, Parkinson’s disease, etc., so I’m not sure it is accurate to use the word “particularly” here.

Line 107: “an” should be “a”

Reviewer 2 Report

In this review, Jihoon Nah give an updated overview of the possible role of alternative mitophagy as potential target for treating heart diseases.

The review is well performed and organized. I have only some suggestions for the author that I think they can be useful to further improve it.

The scope of the review is to demonstrate that alternative mitophagy is a potential therapy against heart disease: i) the section “Therapeutic perspective of alternative autophagy” have only about 30 lines. This is not sufficient. Please greatly improve this section; ii) heart disease is the other topic of the review, but authors only describe two diseases. Why? Please enlarge this section with other cardiovascular diseases. iii) therapeutic strategies reported only target autophagy. Where is the link with mitophagy and, in particular, with alternative mitophagy.

If authors cannot improve these parts, I suggest them to change title, abstract and the overall organization of the review.

Other important points that should be addressed are:

The review focuses on mitophagy, the autophagic removal of mitochondria. Please insert a section to give a general overview of the mitochondrial compartment. This can be useful for a reader that is not expert in the field.

The review focus on mitophagy. I encourage authors to summarize the involvement of this pathway in human diseases, such as the most common neurodegenerative diseases (33079262, 34099564, 30733118), cancer (31121959), Inflammation (31315034), only to cite a few.
